# Comparison of Near-Infrared Spectroscopy-Based Cerebral Autoregulatory Indices in Extremely Low Birth Weight Infants

**DOI:** 10.3390/children10081361

**Published:** 2023-08-09

**Authors:** Howard Chao, Sebastian Acosta, Craig Rusin, Christopher Rhee

**Affiliations:** 1Department of Pediatrics, Division of Neonatology, Baylor College of Medicine, Texas Children’s Hospital, Houston, TX 77030, USA; 2Department of Pediatrics, Division of Cardiology, Baylor College of Medicine, Texas Children’s Hospital, Houston, TX 77030, USA

**Keywords:** cerebral autoregulation, cerebrovascular vasoreactivity, cerebral hemodynamics, near-infrared spectroscopy, premature infants

## Abstract

Background: Premature infants are born with immature cerebral autoregulation function and are vulnerable to pressure passive cerebral circulation and subsequent brain injury. Measurements derived from near-infrared spectroscopy (NIRS) have enabled continuous assessment of cerebral vasoreactivity. Although NIRS has enabled a growing field of research, the lack of clear standardization in the field remains problematic. A major limitation of current literature is the absence of a comparative analysis of the different methodologies. Objectives: To determine the relationship between NIRS-derived continuous indices of cerebral autoregulation in a cohort of extremely low birth weight (ELBW) infants. Methods: Premature infants of birth weight 401–1000 g were studied during the first 72 h of life. The cerebral oximetry index (COx), hemoglobin volume index (HVx), and tissue oxygenation heart rate reactivity index (TOHRx) were simultaneously calculated. The relationship between each of the indices was assessed with Pearson correlation. Results: Fifty-eight infants with a median gestational age of 25.8 weeks and a median birth weight of 738 g were included. Intraventricular hemorrhage (IVH) was detected in 33% of individuals. COx and HVx demonstrated the highest degree of correlation, although the relationship was moderate at best (r = 0.543, *p* < 0.001). No correlation was found either between COx and TOHRx (r = 0.318, *p* < 0.015) or between HVx and TOHRx (r = 0.287, *p* < 0.029). No significant differences in these relationships were found with respect to IVH and no IVH in subgroup analysis. Conclusions: COx, HVx, and TOHRx are not numerically equivalent. Caution must be applied when interpreting or comparing results based on different methodologies for measuring cerebral autoregulation. Uniformity regarding data acquisition and analytical methodology are needed to firmly establish a gold standard for neonatal cerebral autoregulation monitoring.

## 1. Background

Brain injury is a major cause of morbidity and lifelong neurodevelopmental impairment for premature infants. The early diagnosis of severe intraventricular hemorrhage (IVH) has been independently associated with mortality in the extremely low birth weight (ELBW) population [1]. Although the mechanism of brain injury is complex and multifactorial, repeated episodes of hypo-perfusion and re-perfusion have been implicated in the pathogenesis of IVH and white matter injury [2,3,4,5]. The mature brain is protected against fluctuations in cerebral blood flow (CBF) via dynamic changes in cerebrovascular tone and resistance, a process known as cerebrovascular autoregulation. Premature infants are born with limited autoregulatory function and are thus at risk of having pressure passive cerebral circulation, whereby CBF is passive to arterial blood pressure (ABP). Autoregulatory capacity has been shown to progressively improve between 23 and 33 weeks of gestation, in conjunction with the continued development of cerebral vasculature and arteriolar smooth muscle [4]. Until this maturation process is complete, the preterm brain continues to remain particularly vulnerable to both hypo-perfusion and hyperemic mediated injury and subsequent long-term neurodevelopmental impairment.

Despite advances in neonatal care and neuroprotective management strategies, the consequences of prematurity on neurological development remain an important issue within neonatal neurology. Current approaches for preventing cerebral injury emphasize the importance of maintaining normotension to ensure adequate cerebral perfusion. However, within the ELBW population, the definition of hypotension or hypertension remains uncertain. Shifting management strategies to target individualized autoregulatory windows may ultimately reduce the burden of cerebral pathology. This novel approach to hemodynamic management represents an innovative shift away from using traditional blood pressure norms. Individualizing blood pressure goals may reduce overtreatment for hypotension in infants with intact autoregulation and may identify otherwise asymptomatic infants in a high-risk dysregulated state.

Within adult and pediatric literature, more than 20 unique cerebral autoregulation indices have been reported [6]. However, many of these indices cannot be applied to premature infants as invasive intracranial pressure monitoring is generally neither feasible nor practical for this population. Despite such diversity in assessing autoregulation, the fundamental principle remains the same across all methodologies. Each derived proxy for CBF may be quantitatively related to changes in ABP to generate an index of autoregulatory function.

Non-invasive surrogates for cerebral blood flow (or volume) have been studied to quantify autoregulation in the neonatal population. Historically, flow velocity through a large cerebral artery (such as the middle cerebral artery) as measured by transcranial Doppler ultrasound has been utilized. However, there are practical challenges with obtaining long and stable recordings at bedside, and the translation from velocity to flow necessitates the assumption of a fixed vessel dimeter. The advent of near-infrared spectroscopy (NIRS) has dramatically improved the ability to quantify cerebral hemodynamics safely for premature infants over an extended timeframe. However, the methods currently utilized in research encompass many different NIRS parameters and cerebrovascular autoregulation calculations [7]. Currently, there is no consensus regarding the optimal mode of data acquisition, technique for analytics, or timing of data collection. It has been demonstrated within neonatal studies that the details of autoregulation quantification, such as epoch length and the overlapping data percentage, influence the final autoregulation calculations [8,9]. This heterogeneity limits the ability for clinicians to make comparisons between different studies and to draw meaningful conclusions regarding the clinical benefit of autoregulation monitoring. Given this heterogeneity, there is a need for a head-to-head evaluation of the different methodologies, ideally performed within the same cohort, to identify the performance and limitations of each approach. To our knowledge, this is the first comparative study of its kind within the neonatal population.

## 2. Materials and Methods

This is a comparative study based on a re-analysis of prospectively collected data from infants born between June 2016 and July 2021 admitted to the Neonatal Intensive Care Unit (NICU) of Texas Children’s Hospital (Houston, TX, USA). Approval was obtained by the Institutional Review Board at Baylor College of Medicine for the initial study period, and informed consent was obtained from the parents/legal guardians prior to enrollment. The inclusion criteria for the initial studies were premature infants of birth weight 401–1000 g who had an umbilical artery catheter (UAC) in place for clinical reasons. Infants with major complex, congenital anomalies, hydrops fetalis, poor skin integrity, and those in extremis were excluded. The attending neonatologist made all management decisions, including the need for ventilator support, treatment for hypotension, and assessment and treatment of patent ductus arteriosus. Echocardiogram was not routinely available for all infants to assess for hemodynamically significant patent ductus arteriosus.

A cranial ultrasound was performed upon enrollment to assess for early and preexisting brain injury, typically within the first 12 h of life. Subsequent ultrasound studies were performed according to the standard of care for premature infants admitted to the NICU, usually between day of life 5 and 7. A blinded radiologist interpreted all results using Papile et al.’s staging methods [10]. The highest grade of IVH was used for analysis. Data on mortality were included for infants who died during the study period.

### 2.1. Physiological Data Processing

All infants received measurement of ABP with an umbilical artery catheter placed for clinical monitoring. Regional cerebral oxygen saturation (rSO_2_) and optical density were recorded using NIRS (Somanetics INVOS, Medtronics, Minneapolis, MN, USA). Optical density is used in determining the relative tissue hemoglobin (rTHb). All physiological data produced by bedside monitors (GE Carescape B850, GE Healthcare, Chicago, IL, USA) including NIRS were captured in real-time using the Sickbay platform (Medical Informatics Corp., Houston, TX, USA). Only data from the first 72 h of life were included in this analysis. Physiological data were analyzed for missing data (disconnections, movement of probes, etc.) and artifact removal (umbilical arterial line sampling, infants being handled, or spontaneous movements). Each signal was passed through a bandpass filter at a frequency of 0.004–0.05 Hz to isolate the slow wave changes which occur during autoregulatory vasoreactivity [11]. This removes components associated with pulse and respiratory activities while retaining waves in the low and very low frequency range, thus focusing analysis entirely on the cerebrovascular and systemic regulatory responses. Post-acquisition processing of all signals was conducted using MATLAB software (MathWorks, Natick, MA, USA).

### 2.2. Autoregulation Metrics

The moving correlation method for computing autoregulation and cerebrovascular reactivity indices originate from the adult neurosurgical unit at Cambridge University and are well described in the literature [12]. For this study, autoregulation was quantified using three methods, as a moving correlation coefficient between: (1) mean ABP (MABP) and rSO_2_ (cerebral oximetry index, COx); (2) MABP and rTHb (hemoglobin volume index, HVx); and (3) heart rate (HR) and rSO_2_ (tissue oxygenation heart rate reactivity index, TOHRx). The moving correlation analysis generates a time series with values ranging between −1 and 1. A negative index value, or one that approaches zero, is indicative of functional autoregulation and intact pressure reactivity. As autoregulation is lost during a state of pressure passivity, the calculated index will become increasingly positive [4,13].

An example will be provided for COx, which is the continuous moving correlation coefficient between rSO_2_ and MABP. Mean 10 s values of rSO_2_ were compared with average values of MABP over the same duration. Averages over 10 s are used to suppress the influence of the pulse and respiratory frequency wave components. Moving correlation coefficients are calculated between 30 consecutive samples, updated at 60 s intervals from overlapping 300 s epochs. Limiting each epoch to 300 s amplifies the contribution of spontaneous slow wave activity relevant to autoregulation while diminishing effects on CBF caused by changes in arterial carbon dioxide tension, transfusions, fluid boluses, and temperature changes [4]. Each of the other indices included in this study were derived in a similar fashion between their respective input and output parameters.

The autoregulation indices were calculated as minute-by-minute time series for each patient. The subsequent analytic techniques were repeated for a range of temporal resolutions of data, including point-by-point, 8 h moving averages (non-overlapping), and overall mean for each patient.

### 2.3. Statistical Analysis

All statistical and regression analyses were performed using MATLAB version R2021a. Data are reported as means with SD or medians with IQR when appropriate. Data distribution was assessed for normality using the Shapiro–Wilk test. Depending on the distribution, either parametric or nonparametric tests were used. Comparisons of demographic and perinatal variables between groups were performed using an independent Student *t* test, χ^2^, or Mann–Whitney test. The alpha for statistical significance was set at 0.05 for all results describing a *p* value.

Pearson correlations were used to assess the relationships between each pairing of autoregulation indices (COx and TOHRx, HVx and TOHRx, COx and HVx). This analysis was conducted after first performing a Fisher transformation to normalize the correlation coefficient distributions. This was the only instance in which transformed data were utilized within this analysis.

## 3. Results

### 3.1. Infant Characteristics

Fifty-eight infants with a median gestational age of 25.8 weeks and median birth weight of 738 g were included. The characteristics of our patient population are detailed in Table 1. Within our cohort, IVH was detected in 19 (33%) infants, of which 12 (21%) were of low grade (Grade 1 or 2) and 7 (12%) were high grade (Grade 3 or 4). Three infants died during the first week of life.

### 3.2. Data Quality and Availability

A total of 1.5 billion individual data measurements from the 58 infants were collected and processed for this study. The mean age at initiation of NIRS monitoring was 8.4 h of life (range of 2.9–17.6 h of life). The average recording length was 47.7 h (range of 14.6–67.4 h) per individual infant, which was not different between infants with and without severe IVH. Global data availability across the entire study population was greater than 70% between 12 and 46 h of life (Figure 1). Data acquisition was terminated prior to 72 HOL for 12 infants secondary to removal of UAC (n = 8), withdrawal from research study (n = 2), and death (n = 2). Data loss secondary to poor quality, artifacts, or noise averaged 18.8% per patient (range from 1.1 to 71.5%). Significant data loss, defined as greater than 30% of the recording period, was present for 13 patients.

### 3.3. Inter-Index Correlation

The Pearson correlations are depicted in Figure 2. The point-by-point analysis overall demonstrated poor correlations for all autoregulation index pairings. Generally, the Pearson correlation coefficients increased with longer temporal resolutions. Within our study cohort, COx and HVx demonstrated the strongest correlations, although this relationship was at best moderate in strength (range from 0.281 to 0.543 with the greatest value occurring with the patient grand mean set). TOHRx correlated weakly with both COx and HVx. No significant differences in these relationships were found with non-severe and severe IVH subgroup analysis and are not shown here.

## 4. Discussion

NIRS has become an attractive tool for non-invasive and continuous real-time monitoring of tissue oxygenation. Within the neonatal population, NIRS devices have most commonly been utilized to assess the cerebral, renal, and splanchnic circulations. Modern commercial devices are portable and may be easily applied at the bedside, making it well-suited for both clinical and research applications in neonates. Light from the near-infrared range (700–100 nm wavelength) penetrates through skin, bone, and connective tissue and interacts with hemoglobin to determine the percent hemoglobin saturation within tissue. Oxyhemoglobin and deoxyhemoglobin exhibit different absorption spectra within the near-infrared spectrum. NIRS devices utilize at least two different wavelengths of light to estimate the relative concentration of oxygenated and deoxygenated hemoglobin using the modified Beer–Lambert law. Unlike pulse oximetry, which provides a measurement of arterial oxygen saturation, NIRS measurements are not pulse synchronized and can assess the hemoglobin content within arterial, venous, and capillary beds. This results in a composite measure of regional oxygen saturation and reflects the balance between local tissue oxygen supply and demand [6,7,8,9].

Cerebrovascular reactivity is the ability for cerebral vascular smooth muscle to alter basal tone in response to local or central stimuli, such as ABP, arterial oxygen and carbon dioxide levels, sympathetic activation, and effects of neurovascular coupling [14,15,16,17]. When the capacity for vascular reactivity is impaired or exhausted, CBF becomes passive to cerebral perfusion pressure. As there is no device capable of directly measuring CBF, surrogate signals must be used to measure cerebrovascular reactivity.

The rSO_2_ and rTHb are both NIRS-derived parameters capable of assessing changes in CBF. The rSO_2_ represents an arterial-venous weighted estimation of regional oxygenation saturation, which itself is influenced by the cerebral circulation, oxygen content, and oxygen extraction [15,17,18]. Fluctuations in rSO_2_ are thus reflective of changes in CBF under conditions assuming relatively constant cerebral metabolism and oxygen delivery. The rTHb is simply the sum of deoxy- and oxyhemoglobin levels, and can be approximated using the transmittance from a nearly isobestic wavelength of infrared light [11,15]. Its premise as a surrogate for cerebral blood volume assumes that changes in rTHb are directly proportional to changes in cerebral blood volume induced by cerebrovascular vasoconstriction and vasodilation, and are less affected by cerebral oxygenation and changes in metabolism [16]. The use of either of these NIRS parameters to assess for changes in CBF also necessitates the assumption that the regional tissue sampled is representative of the cerebrovasculature at a global level. Further adding to the complexity is the diversity of devices and sensors which are commercially available, with manufacturers often utilizing their own proprietary and device-specific algorithms [7].

Many previous studies have utilized COx or HVx as the method of continuous cerebrovascular reactivity assessment, with TOHRx first introduced by Mitra and colleagues [13] in 2014 as an index of cerebrovascular reactivity uniquely applicable to preterm infants. The majority of these existing studies to date have been limited to relatively small observational studies and are characterized by different methodological approaches. The heterogeneity in monitoring devices, signal filtering and processing, and mathematical models limits the ability to make comparisons between studies and identify a gold standard in assessing cerebral autoregulation.

We examined the relationships between three commonly utilized NIRS-derived autoregulatory indices within the same cohort of ELBW infants. The dataset utilized for this analysis is a robust collection of high-resolution physiological data and one of the largest of its kind in the premature infant population. Our analysis has demonstrated, importantly, that these indices are not closely related to each other, and that it is crucial to understand the basis and limitations for how each index value is derived. Simply, each index may be measuring a different aspect of cerebral hemodynamics and physiology, and that it would not be equivalent to substitute one index for another.

COx and HVx displayed the strongest correlations across all temporal resolutions tested: point-by-point, non-overlapping 8 h moving averages, and patient overall mean. However, this inter-index correlation was only moderate at best, with the highest Pearson coefficient occurring in the patient overall mean set (r = 0.543, *p* < 0.001). The primary difference between these two indices is the proxy utilized for CBF. Our results are of similar magnitude to those reported by Zeiler and colleagues [19] in a comparison between analogous autoregulatory indices in a cohort of adults with traumatic brain injury. Interestingly, the authors of this study also found that most NIRS-based autoregulatory indices displayed weak, or absent, correlation to indices derived from invasive intracranial pressure monitoring.

TOHRx failed to demonstrate strong correlations with either COx or HVx. This finding was confirmed across all temporal resolutions examined. Considering that HVx and TOHRx differ in both the input and output parameters utilized, it is not surprising that these two indices consistently demonstrated the weakest associations in our analysis. It should be noted that TOHRx is not specifically an index of pressure autoregulation, but rather one gauging general CBF regulation and vasoreactivity [3]. For premature infants, cardiac output is regulated primarily through changes in HR due to immature myocardium and an inability to significantly alter stroke volume [2,13,14]. If validated, the use of HR in lieu of MABP as the input signal for assessing cerebrovascular reactivity would preclude the need for invasive arterial catheters and facilitate the ease of monitoring.

For our cohort, it was evident that there was negligible point-to-point correlation between any of the indices (Figure 2A), and that the relationship between indices generally improved across longer temporal resolutions (Figure 2B,C). It may be that there exists a certain temporal threshold (e.g., 8 h of continuous monitoring) that must be satisfied before meaningful conclusions may be drawn regarding the state of cerebral autoregulation. Presently, a minimum or optimal measurement duration has yet to be firmly established in the neonatal literature. Furthermore, the assessment of dynamic autoregulation is strongly dependent on spontaneous fluctuations of the physiological variables of interest. It is plausible that when vitals exhibit minimal variability, as can be encountered during periods of clinical stability, the corresponding autoregulatory indices computed are more a reflection of intrinsic signal noise than clinically relevant physiology. It warrants further exploration if focusing on specific time periods with greater physiological variability would be more sensitive in characterizing the state of cerebrovascular reactivity.

It must be emphasized that the purpose of this study was not to assess the degree to which each index measures, or does not measure, autoregulatory capacity. Additionally, the findings from this study do not invalidate any particular method nor identify one as superior. Rather, we conclude that currently there remains insufficient evidence to prove any one approach is equivalent to another. Current literature utilizing various numerical methods have demonstrated associations between impaired cerebrovascular reactivity and adverse neurological outcomes such as the development of IVH [2,14,15], but with occasionally inconsistent or conflicting results [20,21]. Caution must be applied when interpreting or comparing results based on different methodologies for measuring cerebral autoregulation. For instance, the cutoff value to define the boundary of intact versus impaired autoregulation has been commonly reported between 0.3 and 0.5, but this varies and is unique to each particular index [6,22]. This approach also categorizes autoregulation as a binary phenomenon existing in either an on or off state, whereas the underlying biology and physiology is unlikely to be this simplistic.

Although NIRS has enabled a growing field of research that has provided novel insights into cerebral autoregulation and pathophysiology, the lack of clear standardization in the field remains problematic. While continuous cerebral autoregulation monitoring remains promising and may eventually help guide therapeutic management decisions, there still remain important issues within the realms of data recording, processing, and analysis to address [22,23]. It also remains unclear how to best incorporate variables that may affect autoregulation such as the partial pressure of carbon dioxide, the presence of a hemodynamically significant patent ductus arteriosus, and the effects of medications, such as dopamine and caffeine [14,15]. A common framework is needed to help unify the field and allow for larger multisite collaborations and clinical trials [23].

## 5. Limitations

There are several important limitations to acknowledge in this present study. The inclusion criteria requiring invasive arterial lines may introduce a selection bias as these infants are likely to be more immature and critically ill. Although UAC placement is almost universally attempted for all ELBW infants born within our unit, early line removal (as seen for 8 of our infants within the first 72 h of life) is indicated when no longer necessary for clinical management. The results from our study may not be completely representative of the most stable cohort of infants.

The ability to continuously monitor autoregulatory function requires high-resolution data that are subject to signal artifact, noise, and dropout. While some degree of data loss is practically unavoidable for a study of this nature, the larger fragments of incomplete physiological datasets may introduce another source of unexpected bias into our analysis. There is also no universal and validated method for artifact detection and removal, with some research groups relying on the manual inspection of data and others utilizing automated tools incorporated into specialized software packages (e.g., ICM+^®^, Cambridge Enterprise, UK). In addition to excluding segments of frankly missing data most commonly due to probe disconnections or recording error, we identified and removed transient non-physiological spikes from our analysis (see Table 1).

Based on the timing of cranial ultrasounds, pinpoint identification of the onset of hemorrhage was not possible. Although this study was strictly envisioned as a comparative study evaluating the relationship between several indices of cerebral autoregulation, additional analysis in discrete pre- and post-hemorrhagic states may also have been informative. Impaired cerebral autoregulation has been implicated in the pathogenesis of intraventricular hemorrhage. However, it is also plausible for significant hemorrhage to result in hemodynamic instability and subsequent dysautoregulation. Additionally, periods of hypertension may be deleterious and contribute to the development of hemorrhage, or may instead reflect the need for higher cerebral blood flow in a post-hemorrhagic state [2,7,14]. To address this issue, future studies should be designed to include more frequent cranial ultrasound imaging, for instance, on a daily basis, to more reliably delineate the boundaries between pre- and post-hemorrhagic periods.

Numerous studies in the neonatal population have utilized autoregulation indices to determine an optimal blood pressure (MABP_OPT_) at which autoregulatory capacity is strongest [3,16,24,25]. This analysis is strictly dependent on natural variations in blood pressures to determine the MABP with the lowest corresponding autoregulation index value. While this computation was possible for a small subset of our patient population, the majority of infants did not exhibit a sufficiently broad range of observed blood pressure to fully interrogate the higher and/or lower limits of autoregulation. This precluded our ability to calculate and compare the MABP_OPT_ for each patient using each of the indices in our study.

The relationships, or lack thereof, described in this study are based on a single cohort of premature infants, and should be applied towards generating new hypotheses rather than drawing definitive conclusions. Importantly, insights about the unique underlying physiology of each infant are inevitably lost after averaging into larger groups for data analysis. What appears as an outlier may actually contain important information towards understanding the complex pathophysiology of neurologic injury. Although continuous autoregulation monitoring may be capable of guiding individualized hemodynamic management, this study was not designed to specifically evaluate this potential.

## 6. Conclusions

Cerebral vasoreactivity has been quantified utilizing various input/output parameters and mathematical approaches without clear standardization or large-scale validation. Our study demonstrates that COx, HVx, and TOHRx are not numerically equivalent and that caution must be applied when interpreting or comparing results based on different methodologies. Additional exploration is necessary to better evaluate the performance of each index in detecting pathological conditions and predicting adverse neurologic outcomes. Uniformity regarding data acquisition and analytical methodology are needed to firmly establish a gold standard for neonatal cerebral autoregulation monitoring.

## Figures and Tables

**Figure 1 children-10-01361-f001:**
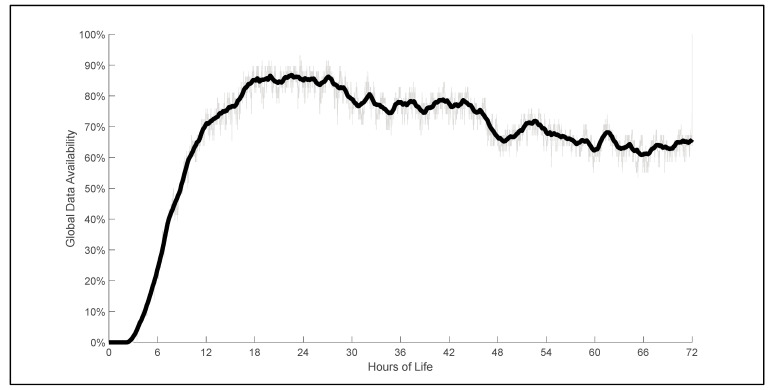
Global data availability versus hour of life across the entire study cohort. Data availability is considered complete when all physiological data of interest (HR, ABP, and NIRS signals) are simultaneously available and free of artifacts or noise. Initiation of NIRS monitoring was most commonly the rate limiting step in data collection.

**Figure 2 children-10-01361-f002:**
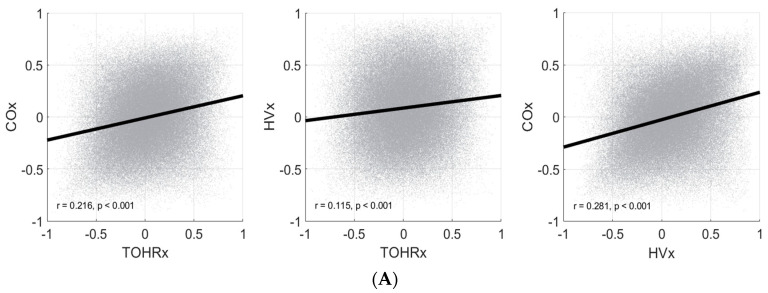
Scatter plots depicting comparisons amongst the autoregulation metrics: COx (derived from rSO_2_ and MABP), HVx (derived from rTHb and MABP), and TOHRx (derived from rSO_2_ and HR). Statistical analysis was performed across a range of temporal resolutions of data, including point-by-point (panel **A**), non-overlapping 8 h moving averages (panel **B**), and patient overall mean average (panel **C**).

**Table 1 children-10-01361-t001:** Characteristics of enrolled infants and data acquisition. Data are presented as mean ± SD or n (%). Reported *p* values reflect the comparison between severe IVH and non-severe IVH subgroups.

	Entire Cohort (n = 58)	Non-Severe IVH (n = 51)	Severe IVH (n = 7)	*p* Value
Patient Characteristics				
Gestational Age, weeks	25.8 ± 1.8	25.9 ± 1.9	24.6 ± 1.1	0.067
Birth Weight, grams	738 ± 135	739 ± 132	734 ± 164	0.925
Male Gender	36 (62%)	32 (63%)	4 (57%)	0.082
Mortality (first week of life)	3 (5%)	3 (6%)	0 (0%)	0.510
Data Acquisition				
Recording length, hrs	47.7 ± 14.2	47.3 ± 14.6	50.3 ± 11.4	0.606
Initiation of NIRS monitoring, HOL	8.4 ± 3.7	8.9 ± 3.6	5.3 ± 1.6	0.012
Data excluded per patient, %	18.8 ± 17.9	18.6 ± 17.9	20.3 ± 19.4	0.786
Patients with <70% data availability	13 (22%)	12 (24%)	1 (14%)	0.582
Patients with >90% data availability	23 (40%)	21 (41%)	2 (29%)	0.523

## Data Availability

Additional data may be obtained upon request from the corresponding author.

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
