# Peer review of "Comparison of Near-Infrared Spectroscopy-Based Cerebral Autoregulatory Indices in Extremely Low Birth Weight Infants"

_children, 2023, doi:10.3390/children10081361_

Round 1
Reviewer 1 Report
The manuscript presents relationships between different indices/proxys for cerebral autoregulation, derived from NIRS-data obtained in a single-center cohort of extremely preterm infants.
The manuscript is well written, applies sound methodology, the data clearly presented and their conclusions are not overstated. I believe the manuscript to a valuable contribution to the field.
Regrettably (I think the authors would agree), cerebral NIRS has not lived up to its promises, so far. I have only one small suggestion: Although the authors do not aim to delineate the validity of different NIRS application in general (and its value as a biomarker for cerebral autoregulation in particular), I would find it reasonable given their expertise, to allow some reasoning regarding this matter in the discussion.
Author Response
Thank you kindly for the time and effort you have dedicated to providing feedback. Changes to the manuscript will be incorporated to reflect the suggestions given. I look forward to hearing from you regarding the revision and to respond to any further questions and comments you may have.
Point 1: The manuscript presents relationships between different indices/proxys for cerebral autoregulation, derived from NIRS-data obtained in a single-center cohort of extremely preterm infants.
The manuscript is well written, applies sound methodology, the data clearly presented and their conclusions are not overstated. I believe the manuscript to a valuable contribution to the field.
Regrettably (I think the authors would agree), cerebral NIRS has not lived up to its promises, so far. I have only one small suggestion: Although the authors do not aim to delineate the validity of different NIRS application in general (and its value as a biomarker for cerebral autoregulation in particular), I would find it reasonable given their expertise, to allow some reasoning regarding this matter in the discussion.
Response 1: We hope that our findings will shed needed insight into the (lack of) relationships between the most commonly utilized neonatal cerebrovascular indices. It is agreed that the primary objectives of our manuscript do not include a more detailed exploration of the various NIRS applications in neonates. However, we would be happy to briefly expand upon this topic within the discussion.
Reviewer 2 Report
Could the authors state which statistical software was used to perform the linear regression analysis shown in Figure 1? The linear regression does not seem compatible with R=-0.62 and p=0.01.
Author Response
Thank you kindly for the time and effort you have dedicated to providing feedback. Changes to the manuscript will be incorporated to reflect the suggestions given. I look forward to hearing from you regarding the revision and to respond to any further questions and comments you may have.
Point 1: Could the authors state which statistical software was used to perform the linear regression analysis shown in Figure 1? The linear regression does not seem compatible with R=-0.62 and p=0.01.
Response 1: MATLAB was used to perform all computations, including statistical analysis. Specifically, the corrcoef function was used for linear regression analysis and to compute r and p values. There is no linear regression depicted in Figure 1. The results of R = -0.62 and p = 0.01 also are not reported within our study. Could additional clarification please be provided?
Reviewer 3 Report
I must confess that we also use NIRS in the NICU. It is a very useful tool for brain oxygenation, and there are studies that correlate NIRS with the need for blood transfusions in neonates.
Your paper is very interesting and it shows the limitations of using some specific parameters. I would recommend mentioning what could be the external factors influencing the results of NIRS and what could be improved in NIRS monitoring.
English is fine
Author Response
Thank you kindly for the time and effort you have dedicated to providing feedback. Changes to the manuscript will be incorporated to reflect the suggestions given. I look forward to hearing from you regarding the revision and to respond to any further questions and comments you may have.
Point 1: I must confess that we also use NIRS in the NICU. It is a very useful tool for brain oxygenation, and there are studies that correlate NIRS with the need for blood transfusions in neonates.
Your paper is very interesting and it shows the limitations of using some specific parameters. I would recommend mentioning what could be the external factors influencing the results of NIRS and what could be improved in NIRS monitoring.
Response 1: Thank you for the observation. We will further elaborate on the factors which we have found to limit the application of NIRS to assess cerebral autoregulation and vascular reactivity. First, we have to take into consideration that NIRS is a non-invasive signal that is subject to signal artifact, noise, and data drop out. There are many different approaches – but no standardization – to address error correction in physiologic data collection. Additionally, there are now many available commercial devices, each with its own proprietary and device specific algorithms. The study of autoregulation necessitates detecting changes in cerebral blood flow, which is not directly measured by NIRS devices. A series of assumptions, such constant metabolism and oxygen delivery, must be taken for changes in regional oxygen saturation to accurately reflect changes in cerebral blood flow. It is also necessary to assume that the regional tissue sampled is truly representative of the cerebrovasculature at a global level. As one can see, even before we get to any signal processing, there are already a wide slate of conditions/parameters which must be satisfied for us to believe that the input data is of reasonable quality and reflective of true underlying physiology. Beyond this step, uniformity regarding data acquisition and analytical methodology are needed to firmly establish a gold standard for neonatal cerebral autoregulation monitoring.
Our analysis has demonstrated that these autoregulatory indices are not closely related to another, and that it is critical to understand the basis and limitations for how each index value is derived. Simply, each index may be measuring a different aspect of cerebral hemodynamics and physiology, and that it would not be equivalent to substitute one index for another.